# Exceptional middle latitude electron precipitation detected by balloon observations: implications for atmospheric composition

Irina Mironova[1], Miriam Sinnhuber[2], Galina Bazilevskaya[3], Mark Clilverd[4], Bernd Funke[5], Vladimir Makhmutov[3,6], Eugene Rozanov[1,7], Michelle L. Santee[8], Timofei Sukhodolov[7,9], and Thomas Ulich[10]

[1]St. Petersburg State University, St. Petersburg, Russia
[2]Institute of Meteorology and Climate Research, Karlsruhe Institute of Technology, Karlsruhe, Germany
[3]Lebedev Physical Institute, Russian Academy of Sciences, Moscow, Russia
[4]British Antarctic Survey, Cambridge, United Kingdom
[5]Instituto de Astrofisica de Andalucia, CSIC, Granada, Spain
[6]Moscow Institute of Physics and Technology, Moscow, Russia
[7]PMOD/WRC and IAC ETHZ, Davos, Switzerland
[8]Jet Propulsion Laboratory, California Institute of Technology, Pasadena, CA USA
[9]Institute of Meteorology and Climatology, University of Natural Resources and Life Sciences, Vienna, Austria
[10]Sodankylä Geophysical Observatory, Sodankylä, Finland

**Correspondence:** Miriam Sinnhuber (miriam.sinnhuber@kit.edu) and Timofei Sukhodolov (timofei.sukhodolov@pmodwrc.ch)

**Abstract.** Energetic particle precipitation leads to ionization in the Earth's atmosphere, initiating the formation of active chemical species which destroy ozone and have the potential to impact atmospheric composition and dynamics down to the troposphere. We report on one exceptionally strong high-energy electron precipitation event detected by balloon measurements in geomagnetic midlatitudes on 14 December 2009 with ionization rates locally comparable to strong solar proton events. This electron precipitation was possibly caused by wave-particle interactions in the slot region between the inner and outer radiation belts, connected with still not well understood natural phenomena in the magnetosphere. Satellite observations of odd nitrogen and nitric acid are consistent with wide-spread electron precipitation into magnetic midlatitudes. Simulations with a 3D chemistry-climate model indicate almost complete destruction of ozone in the upper mesosphere over the region where high-energy electron precipitation occurred. Such an extraordinary type of energetic particle precipitation can have major implications for the atmosphere, and their frequency and strength should be carefully studied.

## 1 Introduction

Energetic particle precipitation into the atmosphere initiates a chain of reactions starting with atmospheric ionization, leading to large changes in middle atmosphere composition, including the formation of hydrogen and nitrogen oxides followed by ozone loss in the stratosphere and mesosphere over ∼30-80 km, and with potential relevance even for tropospheric weather systems and regional climate (e.g. Seppälä et al., 2009; Mironova et al., 2015; Arsenovic et al., 2016; Tsurutani et al., 2016; Sinnhuber and Funke, 2019; Mironova et al., 2021a). Permanent sources of atmospheric ionization are galactic cosmic rays

and solar UV radiation, but the flux of energetic particles can increase by orders of magnitude through episodic precipitation of solar or magnetospheric energetic particles. The precipitation of electrons from the outer radiation belt is a consequence of the violation of the adiabatic motion of the trapped electrons, mostly as a result of the wave-particle interactions (e.g. Millan and Thorne, 2007). Precipitation mainly occurs at high latitudes, in the zone of the auroral oval corresponding to geomagnetic latitudes of ∼65-70° or a McIlwaine parameter of L∼5-6. Comprehensive measurements of mid-latitude electron precipitation from a slot between the outer and inner radiation belts at L∼2-4 have been made on the Van Allen Probes (e.g. Su et al., 2017; Foster et al., 2016). The first results of observations of bremsstrahlung in the atmosphere from precipitating electrons at the geomagnetic latitude 53.9ºN were recently published (Aplin et al., 2021).

Energetic electron precipitation (EEP) leads to the enhancement of odd nitrogen $NO_x$ and odd hydrogen $HO_x$, which play a key role in the ozone balance of the middle atmosphere (e.g. Sinnhuber et al., 2012). The effect of high latitude EEP on atmospheric composition and ozone is confirmed by various observations (e.g. Newnham et al., 2011; Andersson et al., 2014; Newnham et al., 2013; Sinnhuber et al., 2016; Randall et al., 2006) and 3D chemistry-climate models (e.g. Rozanov et al., 2012; Arsenovic et al., 2016; Verronen et al., 2016; Sinnhuber et al., 2018) that account for EEP induced ionization.

Here, we present an exceptional case of high energy electron precipitation (with stratospheric and mesospheric ionization rates locally exceeding those of large solar proton events) from the slot region (2 < L < 4) observed over Moscow (55.96°N, 37.51°E, geomagnetic latitude ∼52°N, L=2.7) on 14 December 2009. To confirm the balloon observations, and to bring those essentially point measurements into a broader context, Polar-orbiting Operational Environmental Satellites and VLF observations are studied as well. Energetic electrons precipitating into the atmosphere decelerate by collisions with the most abundant species. In the middle atmosphere below ∼90 km, these are $N_2$ and $O_2$, which are either ionized or dissociated, starting a complex ion-chemistry reaction chain which ultimately leads to the formation of nitric oxide NO and nitric acid $HNO_3$, see Sinnhuber and Funke (2019) for a recent review. The EEP induced ionization and consequent enhancement of $NO_x$ (N, NO) and $HNO_3$ are confirmed by chemical composition observations from MLS (Waters et al., 2006) and MIPAS (Funke et al., 2014). Model studies with the 1D atmospheric chemistry model ExoTIC (Herbst et al., 2019) and the 3D chemistry-climate model HAMMONIA (Schmidt et al., 2006; Meraner et al., 2016) using the ionization rates derived from the balloon observations demonstrate formation and loss rates of a wide range of neutral species and ozone in the upper mesosphere.

## 2 Observations of middle latitude electron precipitation

### 2.1 Local view: Balloon observations

Balloon observations of energetic particle precipitation in the atmosphere are an important independent source of information for the evaluation of satellite-observed particle flux and energy used in chemistry-climate models, extending the useful energy range from hundreds of keV to several MeV.

The balloon measurements are performed by the radiosonde lifted up to the heights of 30-35 km and returning information on the ionizing particle fluxes at different levels of the atmosphere. The radiosonde sensor consists of two Geiger-Müller tubes arranged as a telescope with a 2 mm Al interlayer between the tubes (Fig. 1b). The device returns the count rates of the upper single tube and the telescope. The single tube is sensitive to X-rays and charged particles (electrons, protons, and muons), while a telescope measures only energetic charged particles but does not respond to the X-ray flux by the atmosphere. During quiet conditions, the radiosonde records the fluxes of secondary cosmic rays. Precipitating electrons are absorbed at altitudes above 50 km, but they generate X-rays via bremsstrahlung, which penetrates into the atmosphere down to altitudes of ∼20 km and can be recorded only by single tube. Intrusion into the atmosphere of solar particles causes count rate enhancement both of a single tube and the telescope, which enables us to distinguish between solar proton and magnetospheric electron precipitation. In the case of smooth growth of the Geiger-Müller tube count rates with altitude, we assume that it is caused by X-ray absorption in air rather than by temporal variations of X-ray flux. Taking the data of a previous balloon flight in quiet conditions as the background and subtracting it from the data of the flight that observed precipitating electrons, we get the X-ray flux vs. atmospheric pressure. The method of evaluation of the energy spectrum of electrons impinging on the atmosphere from the X-ray flux absorption in air was developed on the basis of the GEANT 4 simulation (Makhmutov et al., 2016).

In this study we use observations from the balloon experiment performed by the Lebedev Physical Institute (LPI) every few days since 1957 (Stozhkov et al., 2009), which has so far recorded 589 EEP events at polar latitudes, L=∼5.5, over 1961-2019 (Makhmutov et al., 2016; Mironova et al., 2019a, b; Bazilevskaya et al., 2020), and is complemented by regular balloon launches at midlatitudes. Observations of EEP events in midlatitudes are very rare. However, several candidates have been found since the beginning of the 2000s which have not been studied properly yet. Here we present the most outstanding EEP event recorded in the Moscow region so far.

Data from the LPI balloon observation at 13:26-13:45 UT on 14 December 2009 (Fig. 1c, curve 1) demonstrate a substantial enhancement in the count rate of the single Geiger-Müller tube above ∼20 km (residual pressure ∼55 hPa). This count rate increase of the single Geiger-Müller tube was due to X-ray bremsstrahlung generated by precipitating electrons in the atmosphere at altitudes above 50 km. Note that X-ray radiation from the Sun does not penetrate to heights below 90 km (Mironova et al., 2015). A typical quiet-day result derived on 11 December shows only background in the count rate of the single Geiger-Müller tube (Fig. 1 c, curve 2). The telescopes (Fig. 1 c, curves 3 and 4), which are not sensitive to X-rays, recorded the background due to secondary cosmic rays, confirming particle precipitation as the source of the single tube count rate increase. The precipitation of electrons is characterized by strong variations, sometimes on a scale of several minutes (Mironova et al., 2019b, a). The spatial dimensions of the precipitation area are poorly understood, but can be on the order of hundreds of kilometers (Millan and Thorne, 2007). Therefore, we checked whether the precipitations were observed on December 14 at high latitudes both from observations on balloons and and on satellites. At the polar station Apatity (67.57°N, 33.56°E, L=5.3), a radiosonde was aloft ∼5 hours before the Moscow observation and did not observe enhanced electron precipitation.

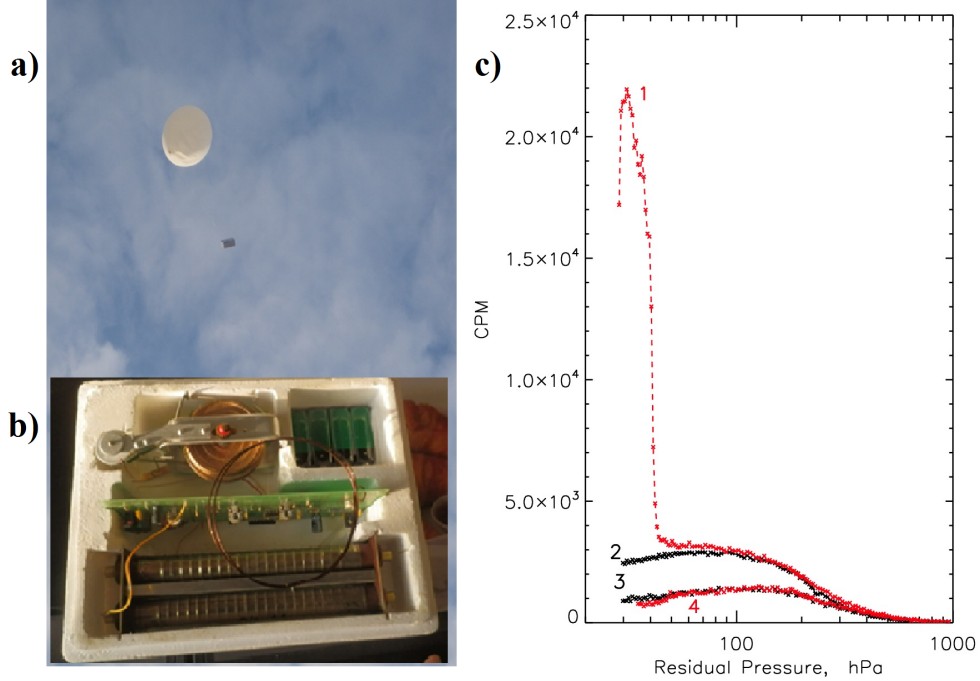

**Figure 1.** Left panel shows a balloon in flight a), and b) a radiosonde layout showing two Geiger-Müller tubes (a top counter referred to as "a single tube" and "telescope" arranged as a telescope detecting particles passing through both tubes and the filter between). Right panel c) shows the results (CPM – count rates per minute of two Geiger-Müller tubes) of observations in the Moscow region on 14 December 2009 (curves 1, 4) and on 11 December 2009 (curves 2, 3). Curves 1 and 2 are the single Geiger-Müller tube count rates, sensitive to X rays. The telescope (sensitive to charged particles) count rate (curves 3 and 4) is multiplied by 3.

## 2.2 Regional view: POES observations

NOAA's Polar-orbiting Operational Environmental Satellites (POES) carry a suite of instruments that measure the flux of energetic protons and electrons at the altitude of the satellite. The Medium Energy Proton and Electron Detector (MEPED) onboard POES consists of telescopes pointing close to zenith (0°) and in the horizontal plane (90°). At polar latitudes, where most precipitation occurs, the vertical telescope registers the precipitated particles, and the horizontal telescope, trapped in the radiation belt. However, the pitch-angle distribution of particles precipitating at midlatitudes has not been studied. Moreover,

the POES angular response functions are not fully understood (Selesnick et al., 2020). We have examined the POES data around the Moscow and Apatity regions for December 2009 with the following limitations: (i) McIllwain parameter L = 2-3, foot-of-field-line latitudes Flat = 52-60°N, foot-of-field-line longitudes Flon = 30-55°, and (ii) L < 8, Flat = 60-70°N, and Flon = 30-55°. The >30 keV, >100 keV, and >300 keV electron channels as well as both the horizontal (90°) and vertical (0°) telescopes were checked.

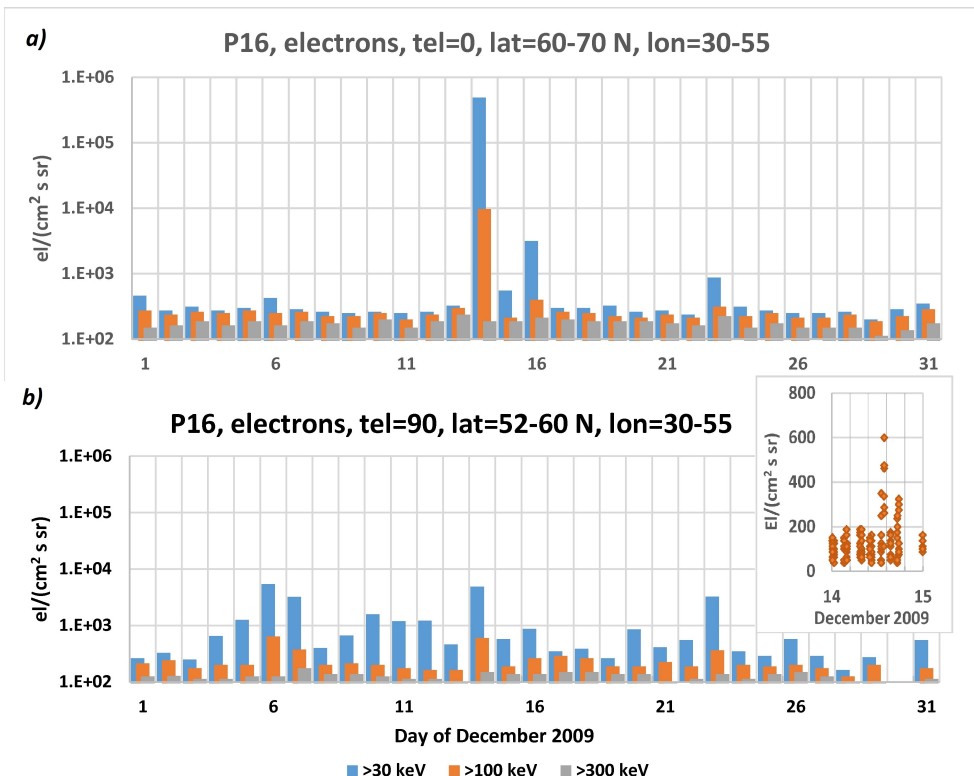

**Figure 2.** Daily data of POES 16 in December 2009. a) Data on electrons recorded by the vertical telescope at the polar latitudes. b) The same but for the horizontal telescope at the midlatitudes, see the Methods section. Inset: same as b) but for 14 December; included data of POES 15 - 18 and MetOp02.

Throughout December 2009, the only significant precipitation of >30 keV and >100 keV electrons in polar latitudes at the longitude of Moscow was recorded on December 14 by the POES 16 vertical telescope, as can be seen in Fig. 2 a). The precipitation was observed at polar latitudes close to the Apatity region at 05:34-05:35 (POES 16), as well as at 07:25-07:26 UT (POES 17) and 13:07-13:08 UT (POES 15). At the midlatitudes there was no electron flux enhancement in the POES 16 100 vertical telescope data at this time, but the horizontal telescope data shows increased particle flux on this day (Fig. 2 b) as well as on 6 and 23 December. Results of POES 15-18 and METOP-02 measurements by horizontal telescopes on 14 December 2009 are given in the insert to Fig. 2 b).

### 2.3   Hemispheric view: VLF observations

Man-made, narrow-band radiowaves are transmitted in the Very Low Frequency (VLF) range from several, mainly mid-105 latitude, locations around the world, particularly in the Northern Hemisphere. The radiowaves can propagate very long dis-

tances subionospherically, and are systematically recorded using a network of VLF receivers known as the Antarctic-Arctic Radiation-belt (Dynamic) Deposition - VLF Atmospheric Research Konsortium (AARDDVARK). Each receiver site is able to log the amplitude and phase of 10 or more transmitter signals, with time resolution of typically 0.1-1 s. Perturbations to the phase and amplitude of the signals are caused by changes in ionization levels at altitudes close to the lower boundary of the D-region (50-85 km). Such perturbations can be caused by energetic particle precipitation (electron or proton), as well as solar flares. Determination of energetic particle precipitation characteristics from VLF perturbation levels requires knowledge of either the flux of particles or the spectrum of energy deposition involved. Knowing one of these parameters allows the other to be calculated. See Rodger et al. (2012) for a detailed description of the calculations required and VLF perturbation responses that are likely to arise.

Subionospheric VLF propagation measurements from several receivers from the AARDDVARK located in the region of Scandinavia (Clilverd et al., 2009) showed a clear burst of precipitation from 13:30-15:00 UT, although the field of view did not include the region around Moscow. The propagation paths impacted by the precipitation spanned the 3<L<8 range, near the geomagnetic latitude of Moscow, although much further west. Figure 3 shows the amplitude variation of the NML transmitter (North Dakota, 25.2 kHz) during 14 December 2009, received at two sites in northern Finland. Both Kilpisjarvi and Sodankylä data show deviations from a representative quiet day curve (QDC) taken from 11 December 2009, with the amplitude difference shown in the right-hand panels. Negative deviations from the QDC can be seen around 05 UT and 08 UT, while a positive deviation is observed after 13 UT. The deviations are consistent with the effect of excess ionisation on the propagation conditions experienced by the subionospheric radio waves at altitudes between 50-90 km (Clilverd et al., 2009).

While solar flares and their ionospheric D-region enhancements are known to reach well below 90 km altitude (Thomson et al., 2005), no M- or X-class flares occurred in December 2009, and there was no flare on December 14, 2009.

## 3  Observations of atmospheric response during the disturbed period

### 3.1  Hemispheric observations: Geomagnetic disturbances

The main driver of energetic electron injection in the Earth environment is the solar wind interaction with Earth's magnetosphere and related geomagnetic disturbances. Electron precipitation at polar latitudes is usually accompanied by enhanced auroral activity indicated by the auroral electrojet AE index, a substantial variation of the disturbance storm time index Dst, and the southward excursion of the Bz component of the interplanetary magnetic field (IMF) (Dungey, 1961). For this reason we took into account the behavior of AE, Dst, and Bz during December 2009. All these hourly-averaged parameters used in our study are collected by the Low-Resolution OMNI data set (King and Papitashvili, 2005).

In December 2009, conditions in the interplanetary space were calm: the magnetic field strength did not exceed 10 nT, the solar wind speed was < 450 km/s. On December 14, the field strength was 6 nT, the solar wind speed was 270 km/s. The

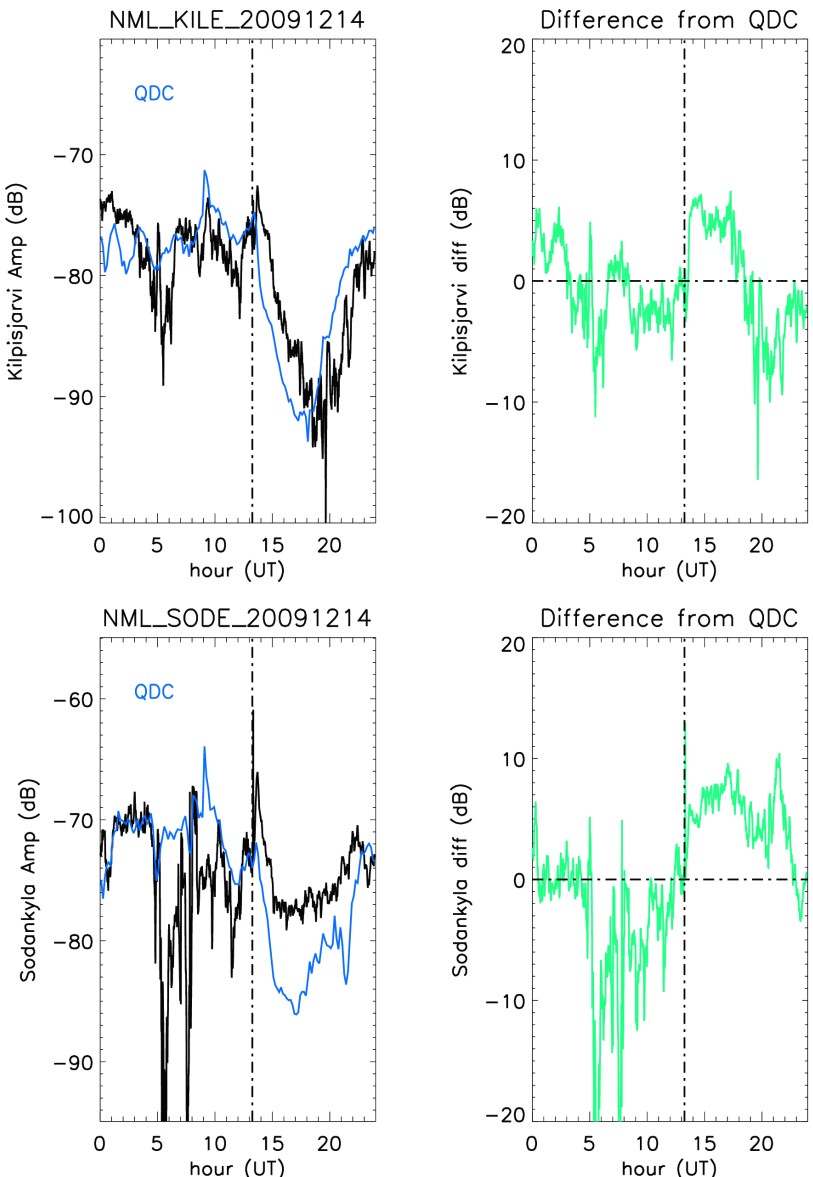

**Figure 3.** The amplitude variation of the NML transmitter located in the northern USA, received at two receivers in Finland on 14 December 2009. Upper two panels show data from Kilpisjarvi, while the lower two panels show data from Sodankyla. A representative quiet day curve for each site (QDC, blue line) is shown using data from 11 December 2009. The difference in amplitude between the signal from 14 December 2009 and the QDC is shown on the right-hand panels. A vertical dashed-dot line indicates the start of a perturbation at 13:15 UT.

general geomagnetic situation in December 2009 was slightly disturbed, as shown in Fig. 4 and Fig. 5a. In the period of
December 13-14, the AE-index reached a value of up to 300 nT (see upper left panel of Fig. 5a, and Dst value -12 nT that can

be considered as a geomagnetic substorm well below the threshold for a geomagnetic storm (Gonzalez et al., 1994; Tsurutani et al., 1990). True, at this time the negative Bz excursion was longest and strongest throughout the month, but not extraordinary. However, a detailed examination reveals features that could lead to the precipitation of electrons at midlatitudes. Fig. 4 presents the geomagnetic activity on 14 December with 1-min resolution. SYM-H index is similar to Dst but with higher time resolution and obtained with more geomagnetic stations. SME is an equivalent to the electrojet index AE, but at 1-min resolution (Bergin et al., 2020).

The event on December 14, 2009, was preceded by two episodes of negative polarity of Bz, which were observed at about 04 UTC and 08 UTC and caused two geomagnetic substorms (Tsurutani and Meng, 1972) with SME index reached up to 500 nT and 350 nT, see Fig. 4. The episodes could lead to the injection of low-energy electrons into the midnight sector of the magnetosphere, the subsequent electron drift to the noon sector, and the generation of chorus waves, which effectively accelerate the electrons to relativistic energies (Horne et al., 2005; Tsurutani et al., 2010, 2016). The precipitation of electrons in the slot region was most likely associated with the interaction with the plasmaspheric hiss usually present in the noon sector, which lead to a rapid scattering of particles into the loss cone (e.g. Abel and Thorne, 1998). This is in agreement with the specific features of the December 14, 2009 event - the fact that it was observed at midlatitudes, in the afternoon sector, and in the absence of strong geomagnetic activity. Such properties correspond to the scenario proposed by Tsurutani et al. (2019).

### 3.2 Hemispheric observations: Satellite observations of trace gases (MLS and MIPAS)

The Earth Observing System (EOS) Microwave Limb Sounder (MLS) (Waters et al., 2006) is an instrument on NASA's Aura satellite, launched in July 2004. MLS observes millimeter and submillimeter-wavelength thermal emission, vertically scanning Earth's limb in the orbit plane from the ground to about 90 km to give daily near-global (82°S–82°N latitude) coverage with ∼15 orbits per day, making measurements during both day and night. Aura is in a sun-synchronous orbit with an ascending (north-going) equator-crossing time of 13:45 local time; it therefore passes the latitude of Moscow shortly after noon local time. One of the important MLS retrieval products that controls stratospheric ozone depletion is nitric acid ($HNO_3$). As $HNO_3$ is a longer-lived reservoir for $NO_x$ and $HO_x$, formation of $HNO_3$ can prolong ozone loss, and also enhance the stratospheric impact by downward transport in polar winter (indirect effect). Here we take into account $HNO_3$ mainly because it is formed very efficiently by ion chemistry reactions, and is therefore a good tracer for particle precipitation impacts. However, due to the relatively poor precision of $HNO_3$ in the upper stratosphere and lower mesosphere, only zonal average data can be used. Here we use the latest version 5 MLS $HNO_3$ measurements (Livesey, N. J., Read, W. G., Wagner, P. A., Froidevaux, L., Lambert, A., Manney, G. L., et al. , 2020).

The Michelson Interferometer for Passive Atmospheric Sounding (MIPAS) on board ENVISAT measured mid-infrared emission spectra in the middle and upper atmosphere during 2002-2012, enabling the retrieval of temperature and a large number of trace species with daily global coverage (Fischer et al., 2008). In this study we use IMK/IAA MIPAS $NO_x$ ($NO_2$, NO) data (version V5R XX 220, XX = NO, $NO_2$) (Funke et al., 2014). ENVISAT was in a sun-synchronous orbit with an equator

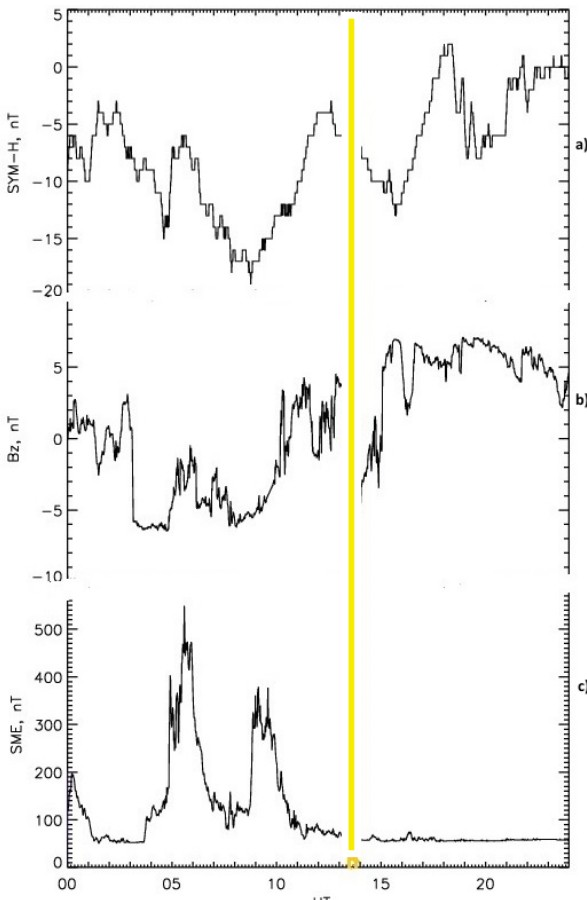

**Figure 4.** Geomagnetic disturbances during December 14, 2009. a) SYM-H index variability. b) IMF Bz variability. c) SME variability. Yellow line marks the time of balloon observation.

crossing time of 10/22 hours local solar time; it therefore passed over Moscow several hours before and several hours after the balloon observation of the electron precipitation on 14 December. Because of the fast horizontal transport in the mesosphere as shown in Figure 5d, a direct observation of the impact of a localized, short-lived event is unlikely.

We analyzed MIPAS $NO_x$ and MLS $HNO_3$ at 68 km altitude at high latitudes (50°-81°N) and at geomagnetic midlatitudes

(10°-55° geomag. lat.), see Figure 5b and 5c. Selection of these latitudinal-longitudinal regions as well as the altitude of the observations was based on balloon energetic electron precipitation observations and HAMMONIA chemistry-climate model results. MIPAS ozone at 68 km shows minimal values below the detection limit of the instrument on December 15 and 24 consistent with the enhanced $NO_x$ and $HNO_3$, but no statistically significant variation of the mean values (not shown). MLS

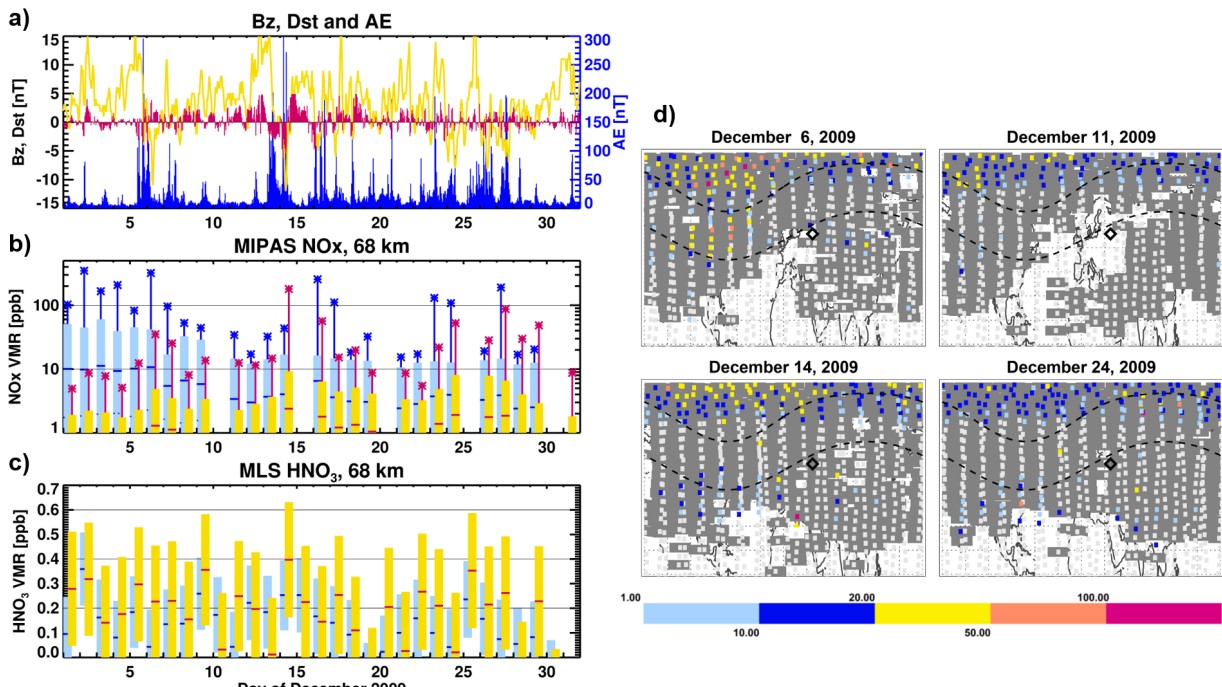

**Figure 5.** a) Bz (red), Dst index (yellow) and AE index (blue). b) MIPAS $NO_x$ (NO+$NO_2$) at 68 km altitude at high latitudes (50°-81°N, blue) and at geomagnetic midlatitudes (10°-55° geomag. lat., red); horizontal line segments mark the mean values, the error bars the 95% percentile, and the stars the maximal values of the day. c) MLS $HNO_3$ at 68 km altitude at high latitudes (50°-81°N, blue) and at geomagnetic midlatitudes (10°-55° geomag. lat., red); horizontal line segments mark the mean values and the error bars the $2\sigma$ standard error of the mean. d) MIPAS $NO_x$ on the satellite overpass footprints on four days (6, 11, 14 and 23 December) at 68 km altitude in the Northern Hemisphere. The dashed lines mark geomagnetic latitudes of 50° and 75°. The diamonds mark the position of Moscow. Colored symbols mark observations larger than the monthly mean plus one standard error. Dark gray areas mark MIPAS footprints where carbon monoxide CO at 70 km is larger than 1.5 ppm.

ozone data is noisier and would have to be averaged over a larger area than the MIPAS observations.

Daily mean values at geomagnetic midlatitudes are highest for both species on 14 December. Given that detection of a localized transient event is unlikely, that an increase in $NO_x$ was observed by MIPAS (and an increase in average $HNO_3$ by MLS) thus suggests either a number of events in different locations on this day, or an event covering a larger area not observable by balloon observations alone. For $NO_x$, maximal values were also much higher than on any other day in December 2009; for

$HNO_3$, a very noisy observation with a large spread, maximal values were not conclusive (not shown). Slightly higher values than on average in either $NO_x$ or $HNO_3$ (or both) were also observed during the periods of negative Bz and substorm activity on 5-6 and 23-27 December. While these enhancements could not be attributed clearly to the location of Moscow and were not

statistically significant, the coincidence in both species with negative Bz and substorm activity suggests electron precipitation into geomagnetic midlatitudes on these days that was strongest on 14 December, see Fig. 5. A closer view of the distribution of enhanced $NO_x$ values on 6, 14 and 24 December (Fig. 5d) shows enhanced values mostly within the auroral oval (over North America) on 6 December, as expected from a period of auroral substorm activity; on 14 and 24 December, the enhanced values occurred mostly southward of the auroral oval in an area reaching from North America over the Atlantic to Northern Europe, with a spread indicating either sporadic precipitation hot-spots at very low latitudes, or fast horizontal transport within the mesospheric polar vortex. The latter possibility is investigated in Fig. 5c by including MIPAS CO observations with vmrs $> 1.5$ ppmv in 70 km altitude as tracers of vortex air, indicating highest NO values at the edge of the area of enhanced CO. 11 December is shown as a reference for a "quiet" day without precipitation.

## 4 Model studies: Potential impact of the mid-latitude electron precipitation event on mesospheric ozone

To estimate the potential impact of the Moscow event on atmospheric composition, we used the 1D atmospheric chemistry model ExoTIC (Herbst et al., 2019) and the 3D chemistry-climate model HAMMONIA (Schmidt et al., 2006; Meraner et al., 2016). Using the ionization rates derived from the balloon observations (Fig. 6a), a model experiment was carried out with the ExoTIC ion chemistry model for the Moscow region, considering ionization from 12-20 UT, to provide formation and loss rates of a wide range of neutral species (Fig. 6b).

### 4.1 Local impact: Ionization rates calculations

Computation of ionization rate ($IR$) requires knowledge of the energy spectra and parameterization of ion production via ionization yield functions. The ionization yield function at the atmospheric depth is the number of ion pairs created by one precipitating electron with the initial energy E at the upper boundary of the atmosphere. The ionization rates (ion pairs $g^{-1}s^{-1}$) are computed as $IR(x) = \int Y(x, E) * F(E) dE$, where $Y(x, E)$ are yield functions, $F(E)$ is a flux of precipitating electrons at the top of atmosphere, x is atmospheric depth and E is energy of the considered particles. The limits of integration are defined by maximum and minimum energy of the considered electrons. During the EEP event observed over Moscow, ionization rates ($IR$) (see Fig. 6a) are computed using a look-up table $Y(x, E)$ with ion production for isotropic flux of precipitating monoenergetic electrons (Mironova et al., 2021b) and electron energy distribution $F(E)$ proposed by balloon-borne observations (see Fig. 1c, curve 1). The background prescribed ionization rates used during December 2009 in the HAMMONIA model are based on the EEP spectra obtained by POES satellites and computed by Atmospheric Ionization Module Osnabruck (AIMOS v1.6, Wissing and Kallenrode (2009)).

### 4.2 Local impact: ExoTIC ion chemistry model results

The Exoplanetary Terrestrial Ion Chemistry model ExoTIC is a 1D stacked-box model of atmospheric neutral and ion composition. It is based on the UBIC model developed for the terrestrial middle atmosphere (Winkler et al., 2009), but has recently been

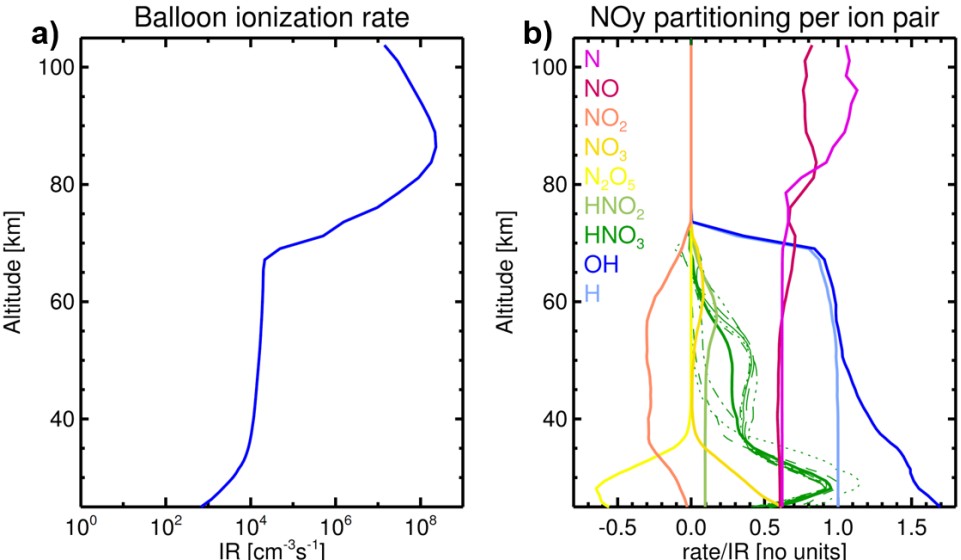

**Figure 6.** a) The atmospheric ionization profile derived from the balloon observations of 14 December 2009 over Moscow. b) Mean formation and loss rates of $NO_y$ species due to ion chemistry calculated hourly from 12:00 - 20:00 UTC over Moscow with the ExoTIC ion chemistry model, showing formation of N, NO, H and OH as well as re-partitioning of $NO_y$ species from $NO_2$ and $N_2O_5$ to $NO_3$, $HNO_2$ and $HNO_3$. For $HNO_3$, individual hourly values are also shown in different line styles to highlight its strong diurnal variability, with distinctly higher values in the 35—60 km region during night-time.

generalized to planetary atmospheres with a wide range of orbital parameters, stellar systems, and base compositions (Herbst
et al., 2019). Temperature, pressure, an initial atmospheric composition and particle impact ionization rates are prescribed externally. The particle energy is distributed to primary ions and excited species based on the atmospheric composition; 60 neutral and 120 charged species are considered, which interact due to neutral, neutral–ion, and ion–ion gas-phase reactions, as well as photolysis and photoelectron attachment and detachment reactions (Sinnhuber et al., 2012). The ion chemistry module called hourly from the base model uses an iterative chemical equilibrium approach and provides formation and loss rates of
all neutral species due to primary ionization, positive and negative ion chemistry which can be used as a parameterization for global chemistry-climate models (Nieder et al., 2014).

The ExoTIC results indicate strong formation of $NO_x$ (N, NO) throughout the middle atmosphere, formation of $HO_x$ (H, OH), and re-partitioning of $NO_y$ species in the altitude range where large positive and negative cluster ions form (below 75
235   km), with strong $HNO_3$ production in the upper stratosphere and lower mesosphere particularly during night-time, see Fig. 6b. These formation and loss rates were provided as input for the HAMMONIA global chemistry-climate model, which does not include a detailed description of the ionospheric D-layer, thus allowing consideration of, e.g., the direct $HNO_3$ production from ion chemistry. As detailed information about the spatial and temporal characteristics of the event comes mainly from

the balloon observations over Moscow, we limited the 3D model experiment to forcing by the available information; i.e., the ionization rates were applied only in the model profile above Moscow and only at the 6 hours prescribed by the balloon and POES observations.

## 4.3 Hemispheric impact: HAMMONIA chemistry-climate model results

The Hamburg Model for the Neutral and Ionized Atmosphere HAMMONIA is a revised version of the general atmospheric circulation model ECHAM5 (Roeckner et al., 2006), in which the upper boundary is raised to $\sim$200-250 km or $1.7e^{-7}$ hPa. A detailed description of the model can be found in Schmidt et al. (2006) and Meraner et al. (2016). The system of hydro-thermodynamic equations in the model is solved by the spectral method with triangular truncation T63, which approximately corresponds to a horizontal resolution of $1.9°\times1.9°$ in latitude and longitude. Vertical resolution is 119 levels. Here we use the model in specified dynamics mode, assimilating ECMWF ERA-Interim data up to 1 hPa. HAMMONIA includes the MOZART3 package to describe atmospheric chemistry (Kinnison et al., 2007). Background ionization rates from auroral and medium-energy electrons and solar protons as well as heavier ions are provided by the Atmospheric Ionization Module Osnabruck AIMOS v1.6 (Wissing and Kallenrode, 2009) with a two-hourly resolution. Ionization effects are described by the 5-ion chemistry scheme in the thermosphere (Kieser, 2011) and by the parameterization of $NO_x$ and $HO_x$ production by energetic particles in the middle atmosphere (Jackman et al., 2005) below $\sim$90 km. Since HAMMONIA does not have a detailed ion chemistry treatment in the ionospheric D layer, the parameterization of Jackman et al. (2005) has been supplemented here by the formation and loss rates of neutral species estimated for the event from the 1D ExoTIC model. Two experiments were conducted with HAMMONIA: with just the background ionization rates from AIMOS and with the background plus the ionization rates estimated from the balloon observations over Moscow on 14 December 2009.

Since we are interested in determining the maximum potential atmospheric impact of the observed midlatitudinal EEP, we estimate the effects with the HAMMONIA model (see Figure 7), applying spatial extreme statistics (global or zonal maximum and minimum values) instead of averaging globally or over a certain region. This approach is justified by the forcing localization and the 3D dynamics of the middle atmosphere, which quickly transports the anomalies induced in chemical species away from their source region. $NO_x$ produced during the event is 1-2 orders of magnitude larger than the unperturbed maximum values in the middle and upper mesosphere above $\sim$60 km (Fig. 7a). The $HNO_3$ mixing ratio in the lower mesosphere reaches values of up to 2 ppbv during the event (Fig. 7b) at 55-68 km, 2-2.5 orders of magnitude larger than the unperturbed maximum values at those altitudes. Modelled $HNO_3$ is additionally plotted at 68 km as maximum zonal values (Fig. 7d) to illustrate the meridional transport of the plume. Transport is mainly defined by the position and shape of the polar vortex, which in the model is an oval with vertices extending to Europe and Alaska (not shown). The initial location of the plume is within the modeled vortex and it circled the globe in about 3 days, after which it got indiscernible. The downward propagation of the odd nitrogen produced by energetic particles in the mesosphere is an important contributor to the stratospheric high-latitude ozone budget (Sinnhuber and Funke, 2019). However, because the event is so localized in the model, this effect is indistinguishable in the global average. The modeled ozone response is caused, therefore, almost completely by the mesospheric $HO_x$ enhancement, leading to the

destruction of as much as 95% of the ozone in the upper mesosphere above 68 km around Moscow during and shortly after the event (Fig. 7c). The negative ozone anomaly quickly disappears in the next few days, but our modelling results suggest that magnitude-wise and in terms of the vertical distribution of the ozone signal the Moscow event is comparable to the solar proton event of January 2005 (Jackman et al., 2011). However, it is not as pronounced in the MLS ozone data as it was in January 2005, suggesting that in our case the precipitation coverage was not as spread as it occurs during usual precipitation events.

## 5 Discussion and Conclusions

On 14 December 2009, surprisingly strong high-energy electron precipitation was observed clearly by mid-latitude balloon measurements. Satellite POES data and VLF receivers confirm these electron precipitations and show that the EEP event extended over a larger area and continued for some time after the observed balloon event, moving northward. Although relatively weak geomagnetic disturbances were observed in December 2009, 2 substorms occurred on December 14 one after another, which could lead to the precipitation of energetic electrons. Mid-latitude energetic electron precipitation can be triggered by wave-particle interactions in the slot region ($2 < L < 4$) between the inner and outer radiation belts. Inside the magnetosphere, the generation of waves called plasmaspheric hiss is especially intense near the plasmasphere boundary (plasmapause). Here the electron scattering dominates the inward radial diffusion, resulting in an "impenetrable barrier" for electrons (Baker et al., 2014; Foster et al., 2016), which precipitate into the atmosphere. It is commonly accepted that the slot between the belts ($L \sim 3$) arises from electron scattering by the waves which can be of either natural or artificial origin initiated by VLF emission of man-made transmitters (e.g. Gombosi et al., 2017; Frolov et al., 2020; Zhao et al., 2019). However, natural or anthropogenic phenomena in the magnetosphere resulting in midlatitude electron precipitation still are not well understood.

The energetic electron precipitation occurred during a period of rather low geomagnetic activity with southward orientation of the interplanetary magnetic field in the near-Earth space. The atmospheric energy deposition during this event was much larger than expected for midlatitude precipitation due to, e.g., hiss forcing, and resembled in strength and altitude coverage large solar proton events. While the perturbations evident in the balloon observations are too short and localized to be directly detectable in the coarser resolution satellite measurements, the hemispheric response of atmospheric species like $NO_x$ and $HNO_3$ are in agreement with a midlatitude precipitation event on this day. Analysis of VLF subionospheric propagation perturbations shows evidence of precipitation during 04-16 UT on 14 December, with several bursts observed within $3<L<8$, including one at the time of the event observed over Moscow.

Both POES and VLF data on 14 December, as well as $NO_x$ and $HNO_3$ observations throughout December, suggest that events indeed lasted for a few hours and covered extended areas during this time and that high-energy electron precipitation can occur even during relatively "quiet" periods without a geomagnetic storm.

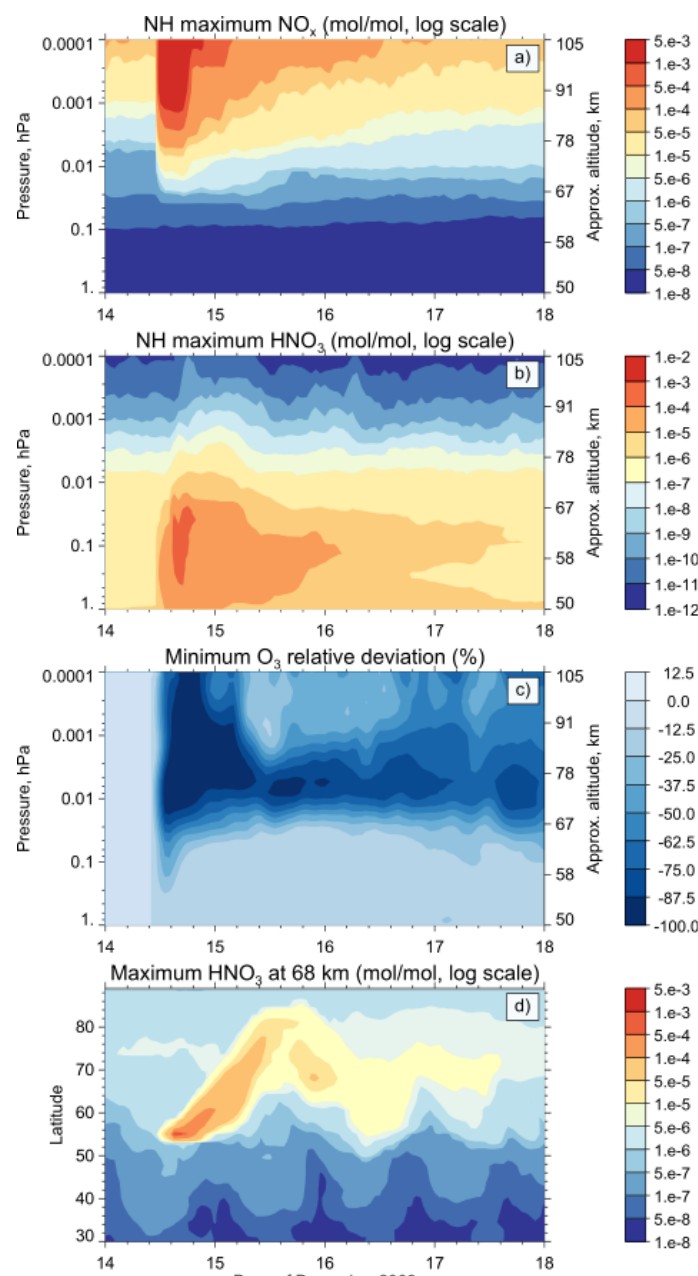

**Figure 7.** HAMMONIA results for a) Northern Hemispheric (NH) maximum value of $NO_x$ volume mixing ratio (VMR). b) NH maximum value of $HNO_3$ VMR. c) NH minimum ozone value of the relative difference between runs with and without the event, and d) zonal maximum value of $HNO_3$ VMR at 68 km.

Satellite observations of $NO_x$ and $HNO_3$ are consistent with precipitation into midlatitudes on several days in December 2009, with the strongest response on 14 December, highlighting that this was an exceptionally strong high-energy electron

precipitation with ionization rates locally comparable to strong solar proton events. In the daily mean average (50°-81°N, 10-55° geomagnetic latitude) the increase is very small and not statistically significant considering the 95% percentile ($NO_x$) or $2\sigma$ standard error ($HNO_3$). However, averaging over large areas / large amounts of data (few hundreds to > 1000 profiles) mutes the maxima, thus increases in hotspots could be much larger. This is indicated in $NO_x$ measurements, where maximal values of about 200 ppb are observed on 14 December compared to a mean value of 2-3 ppb, as well as by the results of the HAMMONIA model experiments. Complete destruction of ozone in the upper mesosphere over the region where high-energy electron precipitation occurred is also shown by HAMMONIA numerical experiments. The predicted ozone losses are largest in the range 68-90 km around Moscow region during and after the EEP event.

The frequency, duration and spatial coverage of these newly discovered electron precipitation events are yet unknown, but results from first model simulations indicate a potentially large impact on atmospheric composition. If such EEP occur frequently and in a larger area over the middle latitudinal region, they could have an accumulated impact much larger than our model results (which assumed only one short, highly localized, event) suggest. Such midlatitudinal EEP events with ionization rates locally comparable to strong solar proton events could be recurrent and have major implications for the atmosphere. Thus, their frequency and strength should be carefully studied.

This conclusion inspires further studies involving a wider network of the balloon-based instruments.

*Code availability.* HAMMONIA chemistry-climate model: code and simulation results can be obtained by contacting TS (timofei.sukhodolov@pmodwrc.ch).

ExoTIC ion chemistry model: code and simulation results can be obtained by contacting MS (miriam.sinnhuber@kit.edu).

*Data availability.* Balloon observations: https://sites.lebedev.ru/ru/sites/DNS_FIAN/479.html

POES: http://www.ngdc.noaa.gov/stp/satellite/poes

OMNI: https://omniweb.gsfc.nasa.gov/form/dx1.html

SYM-H index: http://wdc.kugi.kyoto-u.ac.jp/aeasy/index.html

SuperMAG and SME index: https://supermag.jhuapl.edu/indices/

MLS: https://acdisc.gesdisc.eosdis.nasa.gov/data/Aura_MLS_Level2/ML2HNO3.004/

MIPAS: https://www.imk-asf.kit.edu/english/308.php

AARDDVARK: http://psddb.nerc-bas.ac.uk/data/access/

*Author contributions.*   IM, MS, GB, VM, ER, TS - discussed the idea and wrote the manuscript.

GB and VM -balloon measurements and spectra retrieval.

GB, IM and VM - EEP events selection.

IM – ionization rates calculation.

MS - ExoTIC ion chemistry model: code and simulation results.

ER and TS - HAMMONIA chemistry-climate model: code and simulation results.

Data analysis:

GB – preparation and analysis of POES data, IM and VM – analysis of geomagtetic indexes, MS and BF – preparation and analysis of MIPAS data, IM and MLS – preparation and analysis of MLS data, MC and TU– preparation and analysis of VLF data.

All authors discussed the results and commented on the manuscript.

*Competing interests.*   The authors declare no competing interests.

*Acknowledgements.*   This work is done in the frame of the German-Russian cooperation project "H-EPIC" funded by the Russian Foundation for Basic Research (RFBR project number 20-55-12020) and by the German Research Foundation DFG (grant SI 1088/7-1). Work at the Jet Propulsion Laboratory, California Institute of Technology, was carried out under a contract with the National Aeronautics and Space

Administration. Extreme EEP event selection, ionization rates calculation and numerical experiments with HAMMONIA model was done in the frame of the Russian Science Foundation grant (RSF project number 20-67-46016). Analysis MLS data and numerical experiment results were done in the SPBU "Ozone Layer and Upper Atmosphere Research Laboratory" supported by the Ministry of Science and Higher Education of the Russian Federation under agreement 075-15-2021-583.

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
