# Peer review of "Exceptional middle latitude electron precipitation detected by balloon observations: implications for atmospheric composition"

_Atmospheric Chemistry and Physics, 2021_

## Author Response (AR1)

*Answer Rev1.*

Review of "Exceptional middle latitude electron precipitation detected by balloon observations: implications for atmospheric composition" by Mironova, Sinnhuber, Bazilevskaya, Clilverd, Funke, Makhmutov, Rozanov, Santee and Sukhodolov

The electron precipitation event is quite intriguing and should be published. However contrary to what has been stated in the paper, the interplanetary and geomagnetic conditions are quite ordinary and are not an obvious cause. I will comment in more detail about that. I would say that the event is more of a mystery than one that can be explained. I think the authors will agree with this point of view?

*We are deeply grateful to the reviewer for the very stimulating comments. In fact, there is no scenario in the article that could lead to such unusual precipitation of relativistic electrons. The scenario proposed by the reviewer seems to us the most probable, but it requires a deeper study of related phenomena and is beyond the scope of this article. We would be delighted if the reviewer would join our team to further explore this and other events.*

*We have added suggested information in the text of the paper: All the changes in the text of the paper are highlighted by bold font.*

**Main Comments**

Please provide a blow up of the solar wind parameters and the geomagnetic activity before and after your event. A better geomagnetic index now commonly used is the SME index. It involves many more ground stations and is a better index than AE. The SYM-H index is essentially a one minute average of Dst and therefore with better time resolution is superior to Dst. I recommend that you use this index as well.

*We have added the figure (new Figure 4) with indices SYM-H, Bz, and SME, in the paper and gave corresponding description.*

I have made such a plot to see what was going on in interplanetary space and on the ground for your event. I agree that I have found nothing exceptional for your exceptional electron precipitation event. A ~200 nT substorm is very common and is a weak event. If you look at JGR, 95, A3, 2241-2252, 1990 in their Figure 2, 200 nT is quite weak in comparison to a general distribution. If you look at a distribution of SME values you will find the same results. The same goes for the interplanetary Bz component. A value of ~-5 nT is quite common. Look at the above paper's Figure 1.

*Thank you for calling attention to the paper at JGR, 95, A3, 2241-2252, 1990. We have examined Figures 1 and 2 of the paper. As was expected, the value of geomagnetic indices were quite common and confirmed absence of strong geomagnetic activity.*

Your comment that there is not a magnetic storm is correct. But I take exception to your ascribing your event to a 200 nT substorm. I think this should be toned down since the substorm is so weak.

*We have added into the text that substorm can be considered as a small unexceptional geomagnetic substorm.*

Abstract, line 5. "This event was likely caused...". Perhaps "possibly" rather than "likely"? Soften things a bit. I will later point out that this is a double substorm event. This may possibly be the cause?

*Done. Please see the bold text in the paper.*

The readership should know that your event is one of precipitation of relativistic electrons that are already trapped in the magnetosphere and were accelerated by other mechanisms. The precipitation event does not have anything to do with the acceleration of the electrons to such high energies. This is not clear in the present version of the text. Please correct the text so that the readership will not be misled. Acceleration mechanisms are described in Nature, 437, 227, doi:10.1038/nature03939; JGR, 115, A00F01, 2010. doi:10.1029/2010JA015870; ApJ, 799:39, 2015, doi:10.1088/0004-637X/799/1/39. You may wish to add a short discussion of this for the readership?

*We have added a short discussion. Please see the bold text in the paper.*

**Minor Comments**

Line 17. A good reference to add here is JGRSP, 121, 2016. doi:10.1002/2016JA022499.

*Done.*

Line 117. "AE index larger than 200 nT". In my plot I find two substorms, one with an SME of ~500 nT and a second with ~400 nT. It would be best if you quote exact numbers. And of course show the plot in high resolution.

*Done. We have added a new figure.*

Same line. A Dst of -12 nT is not thought of as a magnetic storm. I suggest that you state this here. A good reference for this is JGR, 99, A4, 5771-5792, 1994. They quote a Dst level which they consider a threshold for a magnetic storm. Most space weather people use this value.

*We have changed the text and added the reference Please see bold text in the paper.*

Lines 120-122. I suggest that you reword this. I see two dips in the IMF Bz. These cause the two substorms as mentioned above. You should refer to this when you show your blow up of the solar wind parameters and geomagnetic activity. It is possible that the TWO substorms may have caused your unusual event. Concerning a good reference for IMF bz and substorms, see JGR, 77, 16, 2964-2970, 1972. The event in Figure 3 has some similarities to your event.

*We have changed the text and added the reference. Please see bold text in the paper.*

Lines 245-6. New results show that plasmaspheric hiss is most intense near the slot region, not near the plasmapause boundary. The new results show that plasmaspheric hiss is quasi-coherent leading to orders of magnitude faster pitch angle scattering. This paper

hypothesizes that the slot is formed much faster than previously believed.   See JGRSP, 124, 10063-10084. 2019. https://doi.org/10.1029/2019JA027102 and references therein.

*Done. Please see bold text in the paper.*

Lines 263-264.  It might be possible that it is not substorm intensity that led to your event, but the double substorm.  For example here is a scenario.  The first southward IMF Bz led to the first substorm through magnetic reconnection (reference Phys. Rev. Lett, 6, 2, 47-48, 1961). This led to the injection of ~10-100 keV electrons in the midnight sector (reference JGR 76, 16, 3587-3611, 1971).  The electrons gradient drift to the noon sector (taking ~ 1 hr see JGR82, 32, 5112-5128, 1977) and there create electromagnetic outer zone chorus through the temperature anisotropy instability (reference JGR, 71, 1, 1-28, 1966).  The chorus propagates into the plasmasphere (reference Nature, 452, 2008. Doi:10.1038/nature06741).   Now a second southward IMF creates a second substorm and the injected 10 to 100 keV electrons drift from the midnight sector to the noon sector.  When these electrons get there they enhance the preexisting hiss in the slot region leading to the parasitic loss of relativistic electrons.

*We have added an information and references.*

*Answer Rev2.*

The authors use a number of observations and simulations to investigate the atmospheric impact of an EEP event identified by a ballon borne experiment. The study is very interesting, but I find that some of the necessary supporting information is not included and it is not always clear why things were done the way they were. Detailed comments and questions are included below and I recommend that these are addressed in a revision before the paper can be accepted for publication.

*We thank the referee for the consideration of the manuscript and useful suggestions that helped to improve the manuscript.*

*We have added suggested information in the text of the paper. All the changes in the text of the paper are highlighted by bold font.*

**Major comments**:

1.     Why is some data not shown? For example, the VLF data is used to justify the extend and occurrence of the EEP event, but is not shown in the paper. Similarly, the POES data is only presented with a daily resolution. Both datasets are used to explain that the event lasted for a few hours and motivate the model simulations. This is even explicitly mentioned in the discussion so it seems important that you present more of these observations.

*We have added figures (VLF: new figure 3, POES data on December 14: inlet of Figure 2) with the required resolution and gave corresponding description.*

2.     With predicted ozone loss at 90% level, I was left wondering why are you not using MLS and MIPAS ozone observations to check for ozone impacts as predicted by the model simulations, as you already use other measurements from both instruments.

*Based on our model results, the predicted ozone losses are largest in the range 68-90 km, but this is not covered by MIPAS data, which in the nominal limb mode only scan up to 68 km. MLS data are available, but noisier and would have to be averaged over a larger area. The large ozone losses are restricted to a small area but not for the zonal averages.*

*MIPAS daily ozone in 68 km, averaged in the same way as MIPAS NOy, is shown in the Figure below, and shows little variation over December; the mean values and percentiles are quite stable over the whole month, with a slight increase in the second part of the month in line with annual variability. The minimum value of December 14 is the lowest value of the month though. However, it is negative, which means it is below the detection limit, and this is also true for December 24, see figure below. While the very low values on December 14 may be considered suggestive, we feel that due to the large overall variability and stable average the values are inconclusive, and shouldn't be included.*

[Figure]

*Figure 1rev: Upper panel:  Bz (red), and AE index (blue). Middle and bottom panels: MIPAS NO$_x$ and O$_3$ at 68 km altitude at geomagnetic midlatitudes (30°-55° geomag. lat., red); horizontal line segments mark the mean values, the error bars the 95% percentile, and the stars the maximal values of the day.*

3.     It is not clear what the reason for using two separate models is. Why do you need to use the HAMMONIA model?  The event is short-lived, and any model results are only presented for a period of four days. Why not use only the ExoTIC ion chemistry model so that the particle impact can be simulated as accurately as possible? This would also mean not needing to account for horizontal transport effects. The balloon measurements are from one point so most accurate ionisation rates are for one location only. Presently only the NOy partitioning from the ion chemistry model are shown. Could you show the density/VMR results from the ExoTIC ion chemistry model?

*As the HAMMONIA results clearly show, horizontal transport is very fast during and after the event in the altitude range affected most. For a probably very localized event like this, horizontal mixing with air masses not affected by the event will also be very efficient. This also means that a one-dimensional model, which can not account for transport and mixing, will overestimate the impact of the event even locally, by a considerable margin. This is different in solar proton events (where one-dimensional models have been used quite successfully in the past), because there, large areas are affected by the particle precipitation, and horizontal mixing, therefore, is not as efficient. For this reason, the ion chemistry model was used to calculate the impact of ion chemistry, but not the evolution of the neutral atmosphere; we feel that the results from HAMMONIA, which include transport and mixing processes, are a more realistic estimate even in the air parcels most affected by the event.*

4.     The HAMMONIA model results in Figure 5 are presented in a format that is difficult to assess (spatial extreme statistics). If the impact is limited to within the polar vortex you could use a vortex average, or present the results as maps for example. Also, since you

have observations to contrast the model results directly to, I suggest presenting the results in a way that allows the reader to make those comparisons.

*As the event is probably restricted to a small area, the impact on the zonal average or vortex average will also be small, consistent with the observations as shown in Figure 5, which show only a small impact on zonally averaged NOx and HNO3, not statistically significant considering the large variability, but very high values locally; we have shown the spatial extreme statistics to highlight this fact. We have changed the figure to make it easier to assess.*

5.      For the ionisation rates, balloon rates and AIMOS rates are used. AIMOS uses POES observations and you show that POES measured enhanced EEP on the 14 December, as also observed by the balloon. What is not clear presently is how this does not lead to "double counting" EEP ionisation in the simulations where AIMOS + balloon rates are used. This need to be clarified. Can you include an example of representative AIMOS ionisation rated in Figure 4a) along with the balloon rates for comparison?

*In the AIMOS model, precipitation within a Kp-dependent prescribed auroral oval is scaled by the observed POES data of this time and geomagnetic latitude / magnetic local time. This means that while the ionization rates within this oval are restricted by observed electron fluxes of this time, the form of the precipitation pattern is not modified in a similar way; it depends on the climatology of POES observations sorted by the Kp index. The observed precipitation occurred over geomagnetic midlatitudes (Moscow), a region outside the auroral oval prescribed with AIMOS, and AIMOS predicted zero precipitation for this region and time. AIMOS data were needed for the background auroral precipitation in the model though.*

*The figure below presents a map of total (electrons, protons and alphas) ionization rates obtained by AIMOS during December 2009. As one can see AIMOS does not show any energetic particle precipitation over the Moscow region and middle latitudes.*

[Figure]

*Figure 2rev: AIMOS ionozation rates (IR) duirng 14 December 2009. Total IR (ip/cm\*\*3/s) from electrons, protons and alpha particles. Maximum values over the vertical direction. Mean values over the day.*

**Further comments:**

- Level of radiation belt information and referencing. For example in the Introduction, line 18: I would not expect most ACP readers to know about adiabatic invariants in the radiation belts. There are no references to help the reader, but, actually, this is not even needed to understand the study (from the atmospheric point of view, the precipitation side is a different story!). Similar situation happens again in the first paragraph of section 5, where you have a lot of detailed information about radiation belt processes that are not really needed in the context. For example, discussing inward radial diffusion does not seem necessary for the results presented in the manuscript, which are focused on the atmospheric impact. I suggest the authors look at these sections critically and revise the text so that it best supports the understanding needed for this work. Whether this is to dial the radiation belt side down, or provide much more context (in which case please include more references) I think is up to the authors.

*The case of energetic electron precipitation that is detectable is not a common case and we have to discuss the reason for the EEP balloon observations over the Moscow region. We have changed the text but maybe we had to add more information on geomagnetic disturbances as was suggested by another reviewer.*

- Since you show data from both, you need explain what the difference and significance between the two POES telescopes (0° and 90°) is.

*Done. We have added the explanation. Please see the bold text in the paper.*

- With the instrumentation relying on X-ray absorption, did you check for any changes in solar X-ray flux during the time of investigation?

*X-ray radiation from the Sun does not penetrate to heights below 90 km. We have added this sentence to the text of the paper.*

- You state that the Apatity balloon didn't observe EEP 5h before the Moscow balloon, but at some point in the three overpasses over Apatity POES (0° telescope) did. Moscow balloon observed EEP, but POES 0° telescope did not, however, POES 90° telescope did. You should explain in the text what the significance of these is. It is not clear at present why the Apatity observations are included, or why the 0° and 90° telescopes provide such different results. The Apatity observations do not seem to be discussed any further in the manuscript.

*We have added the explanation. Please see the bold text in the paper. We hope it is clearer now.*

- Section 3.1. Can you provide some context for the observed Dst and AE values representing moderate levels of geomagnetic activity? These seem rather low values to me. You should also include a reference for southward IMF being needed to enable electron precipitation.

*We have added a short discussion on it and added the reference.*

- Section 3.2 the first paragraph should be moved/merged in the introduction section.

*Done.*

- What do you mean in section 3.2 about HNO₃ controlling stratospheric ozone depletion? Do you mean as a reservoir for NOₓ, in PSCs, or its role in denitrification? It would be good to explain this already in the introduction as you go on to present both observations and simulation of HNO₃. Please include references. As you mention its importance for stratosphere, could you elaborate on the potential contribution of mesospheric HNO₃ (since these are presented) to stratospheric HNO₃?

*As HNO3 is a longer-lived reservoir for NOx, formation of HNO₃ can prolong ozone loss, and also enhance the stratospheric impact by downward transport in polar winter (indirect effect). The role in polar ozone loss is more difficult, as more HNO3 would enhance polar stratospheric clouds (PSC) formation and denitrification; however in spring, when temperatures have risen above PSC thresholds, it would lead to a quicker recovery presumably. However, here it is shown mainly because it is formed very efficiently by ion chemistry reactions, and is therefore a good tracer for particle precipitation impacts.*

- In the context of fast horizontal transport, rather than try to see a localised impact, why not utilise observations of tracers and/or include illustration of the edge of the polar vortex? I understand this could be possible from MIPAS observations.

*We would like to thank the reviewer for this suggestion; we have plotted MIPAS CO at 70 km (see below). CO data seem to indicate a rather extended, persistent vortex throughout December 2009, with most of the enhanced NO values in midlatitudes at the edge of the vortex area. However there appears to be a gap between those values and the enhanced values in the middle of the vortex, and the higher midlatitude values might be due either to outflow from the vortex center, or to local production in an area with already slightly enhanced NO at the vortex edge and quick redistribution at the inner edge of the vortex with the polar night jet. For clarification, we have highlighted areas with CO > 1.5 ppm at 68 km as a threshould of vortex air to the maps in Figure 5.*

[Figure]

***Figure 3rev***: *MIPAS CO at 70 km.*

- For the averaged MIPAS and MLS observations at latitudes 10-55 geomagnetic, geographically this includes the equatorial region. This seems concerning, why include geomagnetic latitudes as low as 10°?

*In Figure 1rev of this document we choose the region 30°-55° geomag. latitude, but it does not change the observation results, so we decided to leave the figure in the paper as is.*

-        Section 3.2, line 157. Substorm activity has not been mentioned before this. Some further explanation is necessary.

*We have discussed the geomagnetic disturbances in the previous section. Here we have added information as a link to the figure so that the reader can understand our explanations.*

-   line 164. What do you mean here by fast horizontal transport? Do you mean that the polar night jet extends to these low latitudes (what about horizontal mixing within the vortex)? If we assume a 70 m/s zonal wind speed, and take a simple zonally symmetric vortex, at latitudes of 40°-50° it would take 4-5 days to circle the Earth. I don't quite see how the hotspot would have move such a large distance within the same day, even if the vortex is likely not zonally symmetric. Perhaps this just need a little bit more clarification. If anything, the MIPAS maps seem to suggest to me that there could have been a precipitation hotspot over the North-America/North-Atlantic sector on 14th Dec (more analysis would be needed to show this of course)!

*Moscow is located at 55° N, which is well within the polar vortex area in the upper stratosphere and the mesosphere. On Figure 4rev we presented the zonal wind speed from Era-Interim data for 15 Dec 2009 and 0.1 hPa. As you can see, the vortex does cover the region around Moscow and the wind speed can reach quite large values up to 92 m/s. Also we directly checked the HNO3 plume produced in the model and it does circle the globe reaching the Moscow longitude in about 3 days.*

[Figure]

Era-Interim u-velocity at 0.1 hPa on 15 Dec 2009

u-velocity (m/s)

-92.0       -55.2       -18.4       18.4       55.2       92.0

Data Min = -52.6, Max = 92.4

***Figure 4rev**: 0.1 hPa zonal wind speed on 15 Dec 2009 from Era-Interim reanalysis data.*

*Concerning the MIPAS observations, MIPAS has a sun-synchronous orbit with 10am/pm local solar time. This unfortunately means that the morning observations of MIPAS passed over Moscow before the event, while the night-time observations passed over presumably after the event, when the plume already had been diluted/moved away; and there are unfortunately no MIPAS observations in the nominal limb mode on December 15. So the MIPAS observations of an apparent hot-spot over Northern America are presumably more likely due to local production; transport from an event in the Moscow area being highly improbable over such a long distance within a few hours. So the point of the MIPAS observations is to highlight that there likely was mid-latitude precipitation on that day, not confined to Moscow only, though the total extent of the event can't be deduced from the MIPAS observations due to the sparse temporal coverage of the sun-synchronous orbit.*

*Concerning the MIPAS results over the other regions, we would be happy to model those regions as well, however unfortunately we don't have any reliable focing information for them, given that the physical mechanism for such a low-latitude precipitation is currently unclear, so that a simple extrapolation of the point Moscow observation would be a too crude estimate. Therefore we decided to focus on what has been measured directly and to see if it can provide a signal in atmospheric chemistry components that is distinguishable from the background.*

- Figure 3 caption: What is the rhombi mark? Diamond?

*Yes.*

- Are the models forced with ionisation for 6h or 8h? There seem to be conflicting numbers in the manuscript. On this topic: the balloon was only up for a very short period of time: 13:26-13:45 UT. How does this justify 6h/8h of forcing? POES observations supporting the use of 6h/8h are not presented.

*The difference in timing is between ExoTIC and HAMMONIA: model runs with ExoTIC covered 8 hours to cover most of the night, but in the end only six hours were used in the HAMMONIA runs. The six hours were taken to be roughly consistent with the period of time enhanced particle fluxes were observed over midlatitudes by the horizontal telescope of POES at all as now shown by the inset in Figure 2. VLF and POES observations show precipitation on that day only in polar latitudes.*

- Is constant ionisation used for the whole 6h/8h? If yes, you should provide some justification for this.

*A constant rate for a certain time is a simplification certainly; but as we know the ionization only for the time of the balloon launch, we have to make some assumption. Note our aim was not to reproduce any observations here, just to highlight the potential of such an event.*

- Relating to one of the major comments: Why use different ionisation rates in the HAMMONIA model? How much do these differ from the balloon ionisation rates? From reading section 4.3, it really seems like there could be double counting of EEP, with AIMOS representing medium energy electron precipitation using POES and adding on the balloon ionisation rates (also representing medium energy electron precipitation!).

*See response to the major comment 5: at the mid-latitude position of Moscow, the AIMOS rates do not show precipitation.*

- I find Figure 5 confusing: These are simply NH maximum of minimum values, without any regional restriction. The contour labels are too small (one can not see the powers clearly).

*We have changed this figure, so that the contours are now more distinguishable from each other. Former Figure 5 (now Figure 7) presents not the maximum of minimum values but the actual hemispheric maximum values for NOx and HNO3, hemispheric minimum values of the ozone signal due to HOx, and the zonal maximum values for HNO3 at 68 km. By doing so, we wanted to highlight that even a localized event can produce values that are larger than the background by several orders of magnitude.*

- Figure 5 and section 4.3: There doesn't seem to be clear evidence for residual circulation in these figures. The timescale of 4 days is not really enough to see vertical transport clearly. I suggest carefully revising the text regarding residual circulation or showing more supporting evidence.

*The reviewer comment is correct. This is the part that was added in the first analysis, which we forgot to exclude later. We have now corrected this part in the text.*

- Line 238: You are not presenting model simulations over Moscow, so you do not show ozone loss over Moscow during the event. Consider the use of the extreme statistics not tied to a specific location, or revise here.

*Correct, we have changed it to the region "around Moscow", since the ozone signal in the upper atmosphere is not exactly above Moscow anymore but is is also moving by fast high-altitude winds. But, even though we are looking at the hemispheric scale, the only difference that was implemented between two runs was the extra particle precipitation over Moscow, therefore we can directly associate the anomalies appearing during the first days after the event with the implemented forcing. We have added some more text into that part to clarify the discussion.*

- Section 5: "*Both POES and VLF data on 14 December, as well as $NO_x$ and $HNO_3$ observations throughout December, suggest that events indeed lasted for a few hours…*" It seems critical that you present the POES and VLF observations in high enough a temporal resolution to support this statement.

*We have added figures with the best time resolution for POES and VLF data.*

- "*Complete destruction of ozone in the upper mesosphere over the region where high-energy electron precipitation occurred is also shown by HAMMONIA numerical experiments.*" Statement like this is conflicting when you present the results not tied to a location. I can see why you many want to present the maximum impact possible (in which case the ExoTIC model may have been sufficient, rather than HAMMONIA). On the other hand as you state, horizontal transport is relative rapid and these effects may be mixed in quickly and in this context also zonal averaged may be justified. I strongly recommend you consider presenting the HAMMONIA results in a different format (perhaps a time series of horizontal maps with a selected time resolution). I would also very much like to see the ExoTIC model results.

*As we have very limited forcing information, our goal was to illustrate the importance of what has been observed in the context of the global background with all its complexity. We couldn't do that with the 1D model as the background is also highly affected by circulation*

*and dynamics. We have edited some parts of the text now, making it clear that we don't want to exaggerate the effects and highlighting that the effects from the observed forcing become indistinguishable from the background after few days. Below we present a map with the ozone anomaly averaged over the mesosphere 14 hours after the start of the event before it starts to get filled up during the daytime by fast photolysis. We could add such a figure to the main text, but we don't think that it will contribute more than it will distract from the main message.*

[Figure]

**Figure 5rev:** *Ozone anomaly averaged over the mesosphere 14 hours after the start of the event (15 January 2 AM UTC).*

- Conclusions "*This conclusion inspires further studies involving a wider network of the balloon-based instruments.*" I think for a commend such as this, it is important to show how different the balloon ionisation rate is to the AIMOS ionisation rates.

*See response to the major comment 5. It is not possible to plot if AIMOS really is zero over Moscow at this time.*

- Please pay attention to the varied used of "*middle latitudes*" and "*midlatitudes*". It is confusing, I suggest using just "midlatitudes".

*Done.*

---

## Author Response (AR2)

Answer to Rev.2 questions:

It's great to see more data added, supporting a much wider precipitation event. Indeed one of the main remaining comments I have is the mixing of the global signals with the emphasis on very localised effects and small precipitation area. It is challenging for the reader to separate between the global responses and the local emphasis. The clarification in the reviewer responses was very helpful in this regard, but the issue remains the manuscript. For example, the POES data is presented in a very local sense (limited lat and long rage), VLF captures wider area, MIPAS and MLS provide a more global view but discussed in context of local impact (the extreme statistics reflect the impact totally elsewhere and not over Moscow), models are used to evaluate local impact, but HAMMONIA in a global sense (including transport). I may just require a careful, small revision of the text to guide the reader. Perhaps using sub-headings to structure the contents in this regard?

We thank the referee for the consideration of the manuscript and useful suggestions that helped to improve the manuscript. Following the reviewers' suggestion, we have changed most of the section headers to make the structure of the paper clearer regarding local (over Moscow), regional (broader region centered around Moscow) and hemispheric impacts. We have also added suggested information in the text of the paper, see response to comments below. All the changes in the text of the paper are highlighted by blue fonts.

The reason for enquiry about solar x-ray flux relates to solar flare activity, which should be excluded as a potential driver for the x-ray observations presented. My understanding is that solar flares can reach well below 90 km altitudes. See: Thomson, N. R., C. J. Rodger, and M. A. Clilverd (2005), Large solar flares and their ionospheric D region enhancements, J. Geophys. Res.,110, A06306, doi:10.1029/2005JA011008
From my checking, there do not appear to be any large flares on 14 Dec 2009

Yes, we also could not find any solar flares class M and X during December 2009 and was no any solar flare during 14 Dec 2009.
We have added in section 2.3:
While solar flares and their ionospheric D-region enhancements are known to reach well below 90 km altitude (Thomson et al., 2005), no M- or X-class flares occurred in December 2009, and there was no flare on December 14, 2009.
.

MIPAS HNO3: Thank you for clarifying the purpose of the use of HNO3. This is an important point, please also explain this in the manuscript text.

We have added:
As HNO3 is a longer-lived reservoir for NOx and HOx, formation of HNO3 can prolong ozone loss, and also enhance the stratospheric impact by downward transport in polar winter (indirect effect). Here we take into account HNO3 mainly because it is formed very efficiently by ion chemistry reactions, and is therefore a good tracer for particle precipitation impacts.

In the revised MIPAS maps CO has been added as a tracer. However, this is not yet described in the revised version in section 3.2 or the figure caption. Does the vortex really extend to Sahara? This contradicts the statement on lines 265-266

Please note that the extension of the vortex in the mid-mesosphere (70 km) as shown here can is different to the stratospheric vortex, and this does not mean that the stratospheric vortex below 45 km extends so far South. The reference in lines 265-266 refers to the vortex in the HAMMONIA model, not to the observations.

We've added the description to the figure caption, and added the following text at the end of section 3.2:
….or fast horizontal transport "within the mesospheric polar vortex. The latter possibility is investigated in Fig. 5 c by including MIPAS CO observations with vmrs > 1.5 ppmv in 70 km altitude as tracers of vortex air, indicating highest NO values at the edge of the area of enhanced CO."
We've added "in the model" to the sentence in (formerly) lines 265-266 for clarification

Section 4 heading change "ozone layer" -> "mesospheric ozone". Ozone layer as a term refers to the primary ozone maximum in the stratosphere.

Done

Please revise "rhombi" -> "diamond"

Done

Line 13: "nitric" -> "nitrogen"

Done

Line 21: This new sentence needs revising, and a reference.

Sentence changes from:
Mid-latitude electron precipitation has received proper observation-only recently. Comprehensive measurements of precipitation from a slot between the outer and inner radiation belts at L~2-4 have been made on the Van Allen Probes (e.g. Su et al., 2017; Foster et al., 2016).
to:
Comprehensive measurements of mid-latitude electron precipitation from a slot between the outer and inner radiation belts at L~2-4 have been made on the Van Allen Probes (e.g. Su et al., 2017; Foster et al., 2016).

Line 91: "captured" -> "trapped"

Done

Lines 314-315: This sentence seems to be missing something, please revise

Sentence changes from:
The predicted ozone losses are largest in the range 68-90 km, but this is not covered by MIPAS data, which in the nominal limb mode only scan up to 68 km. MLS data are available, but noisier and would have to be averaged over a larger area. The large ozone losses are restricted to a small area but not for the zonal averages.
to:
The predicted ozone losses are largest in the range 68-90 km around Moscow region during and after the EEP event.